# INTERPRETABLE NETWORK STRUCTURE FOR MODELING CONTEXTUAL DEPENDENCY

## ABSTRACT

Neural language models have achieved great success in many NLP tasks, to a large extent, due to the ability to capture contextual dependencies among terms in a text. While many efforts have been devoted to *empirically* explain the connection between the network hyperparameters and the ability to represent the contextual dependency, the *theoretical* analysis is relatively insufficient. Inspired by the recent research on the use of tensor space to explain the neural network architecture, we explore the interpretable mechanism for neural language models. Specifically, we define the concept of separation rank in the language modeling process, in order to theoretically measure the degree of contextual dependencies in a sentence. Then, we show that the lower bound of such a separation rank can reveal the quantitative relation between the network structure (e.g. depth/width) and the modeling ability for the contextual dependency. Especially, increasing the depth of the neural network can be more effective to improve the ability of modeling contextual dependency. Therefore, it is important to design an adaptive network to compute the adaptive depth in a task. Inspired by Adaptive Computation Time (ACT), we design an adaptive recurrent network based on the separation rank to model contextual dependency. Experiments on various NLP tasks have verified the proposed theoretical analysis. We also test our adaptive recurrent neural network in the sentence classification task, and the experiments show that it can achieve better results than the traditional bidirectional LSTM.

## 1 INTRODUCTION

Language modeling is a fundamental topic in natural language processing (NLP), and neural network based language models have achieved great success in many NLP tasks (Cheng et al., 2016; Ko et al., 2017; Peters et al., 2018a;b). For instance, ELMo (Peters et al., 2018b), as a contextualized word representation model, significantly improves the results in a broad range of NLP problems. One important reason is that the recurrent network and its variants can capture the long-range/short-range contextual dependencies among terms in a text (Graves, 2013; Yang et al., 2017; Peters et al., 2018a). Researchers are trying to empirically explain the relation between network hyperparameters (such as the number of layers) and the ability to represent contextual dependencies (Karpathy et al., 2015; Peters et al., 2018a). However, the theoretical explanations are relatively insufficient.

Recently, Peters et al. (2018a) investigated the effect of network depth on the range of contextual dependencies represented in neural language models. Empirical evidences show that the bottom layer can capture short-range dependencies, while deep layers are more capable to represent long-range dependencies. It is observed that different NLP tasks require different layers of pre-trained embeddings in (Peters et al., 2018a). Specifically, for the coreference resolution task requiring global syntax structures, long-range dependencies among words should be modeled, and thus embeddings vectors in deep layers are more useful than those in bottom layers. On the other hand, for the named entity recognition task, it requires relatively local syntax to be modeled, and then the embeddings in bottom layers can work well (Peters et al., 2018a). However, while the empirical observations are insightful, it is challenging to theoretically derive the quantitative relation between the neural network structure and the representation ability for contextual dependency.

Our work is inspired by these studies on the relation between neural network and tensor network (Cohen & Shashua, 2016a; Levine et al., 2017). These studies have shown that the correlation between

input regions of an image can be calculated by the separation rank. Based on such a measurement and its analysis, the structure of convolutional neural network can be explained in a tensor space (Cohen & Shashua, 2016a). Levine et al. (2017) further utilize the separation rank to measure the long-term memory capacity and analyze its relation to the network depth, by mapping the recurrent neural network to a tensor network.

In language modeling tasks, tensors have been used for text representation and language modeling (Liu et al., 2005; Socher et al., 2013; Pennington et al., 2014; Qiu & Huang, 2015). Recently, the tensor network (as a method to model high-order tensors) has been adopted in language modeling, and a tensor space language model (TSLM) has been proposed (Zhang et al., 2019). TSLM was proved to be a generalization of the $n$-gram and RNN language models. TSLM bridges the gap between the tensor network and language modeling, and then paves the way for the further theoretical analysis of neural language modeling in tensor space.

In this paper, we aim to provide quantitative analyses for explaining the mechanisms of modeling contextual dependencies in recurrent neural network (RNN) and its variants (e.g., LSTM, EMLo). We first define the concept of separation rank in the language model process, and show that this separation rank can measure the dependencies in a sentence. We further derive the lower bound of the separation rank in the tensor space language model, which is a combination number of network depth and network width, etc.

We then provide a lower bound analysis for the separation rank defined in language modeling. It shows that along with the increasing number of network layers $L$ (representing the network depth), the lower-bound can be increased by an *exponential* level, while increasing the number of hidden units $R$ (representing the network width) can improve the lower-bound by a *linear* level. The lower-bound is associated with the separation rank, which measures the degree of the contextual dependencies.

The method of Adaptive Computation Time (ACT) (Graves, 2016) allows recurrent neural networks to learn how many computational steps (i.e., the depth) to takes on an input. However, this method implies an assumption that the states and outputs are approximately linear, and it lacks a theoretical interpretation. The effect of depth on the ability of neural network to model contextual dependencies is more direct, as mentioned above (Levine et al., 2017; Zia & Razzaq, 2018). For a natural language processing task, it is a challenge task to get an adaptive neural network structure (i.e., the number of layers). Inspired by this work (Graves, 2016), we design a bidirectional adaptive recurrent neural network based on the guidance of the separation rank in language model.

In order to verify our theoretical analysis, we carry out various NLP tasks that reflect different degrees/ranges of contextual dependencies. The experimental results can explain the effect of network depth (represented by network layers) on the contextual dependencies modeled in the neural language models. In addition to the network depth, our experiments show that increasing the network width (i.e., the number of the hidden units) can also increase the modeling ability for both long-range and short-range contextual dependencies. Moreover, increasing the network depth can be more effective than increasing the network width, on the tasks requiring long-range dependency. Furthermore, in order to search a minimum layers and effective modeling contextual dependency, we train the new adaptive neural network on the sentence classification task. The results show that our model achieves better results than a fixed depth recurrent neural network (i.e., BiLSTM).

## 2 PRELIMINARIES

We first briefly introduce the basic language model, as well as the recently developed tensor space language model (TSLM) (Zhang et al., 2019), which is proved to be a generalized approach to both $n-$gram and recurrent language models.

### 2.1 BASIC LANGUAGE MODEL

Given a sentence $s$ of $n$ words denoted as $(w_1, w_2, \ldots, w_n)$, a language model computes the joint probability of a sentence. Its joint probability can be written as:

$$p(w_1^n) = p(w_1, \ldots, w_n). \tag{1}$$

$N$-gram language model computes the joint probability of the sentence by calculating the probability of a word $w_i$ given the history $(w_1, \ldots, w_{i-1})$.

## 2.2 TENSOR SPACE LANGUAGE MODEL

Recently, a Tensor Space Language Model (TSLM) (Zhang et al., 2019) was proposed, which can consider all the combinatorial dependencies among words through the tensor product. Tensor product is a fundamental operator in tensor analysis, denoted by $\otimes$, which can map two low-order tensors to a high-order tensor. It is proved that TSLM is a generalization of both $n$-gram (when the word vectors are represented with *one-hot* vectors) and a recurrent language modeling process (when the word vectors are represented with a low-dimensional dense word vector).

In TSLM, for a sentence $\mathbb{S} = \{w_1, \ldots, w_n\}$ with length $n$, each word $w_i$ can be represented by a $m$-dimension word vector $\boldsymbol{w}_i$ ($i \in \{1, \ldots, n\}$). The basic formulation of TSLM can be written as:

$$p(w_1^n) = \sum_{d_1,\ldots,d_n=1}^{m} \mathcal{T}_{d_1 \ldots d_n} \mathcal{A}_{d_1 \ldots d_n} \tag{2}$$

where $\mathcal{A}_{d_1,\ldots,d_n}$ are the entries from tensor $\mathcal{A}$ which is essentially rank-one tensor, i.e., $\mathcal{A} = \boldsymbol{w}_1 \otimes \ldots \otimes \boldsymbol{w}_n$. $\mathcal{T}_{d_1,\ldots,d_n}$ are entries from tensor $\mathcal{T}$, which is a higher rank tensor. $\mathcal{T}$ can encode the network parameters, e.g., hidden layers matrix and states matrix, after TSLM is reduced to a recurrent language modeling architecture.

## 3 SEPARATION RANK AS MEASUREMENT FOR CONTEXTUAL DEPENDENCY

In this section, our aim is to define the separation rank in TSLM. In Sec. 3.1, we first describe the basic ideas for the contextual dependency and expose the variables which are related to the contextual dependencies. Then, we define the separation rank in tensor space language model in Sec 3.2. For better readability, we put detailed proofs in supplementary appendices.

### 3.1 BASIC IDEAS FOR CONTEXTUAL DEPENDENCY

The separation rank measures how far a function is from being separable (Cohen & Shashua, 2016b). $p(w_1^n)$ in Eq. 1 can be considered as the probability density function, given a sentence $S = (w_1, \ldots, w_n)$, where $n$ is the number of words. Without loss of generality, we can split a sentence into two disjoint context subsets, $S_1 = (w_1, \ldots, w_i)$ and $S_2 = (w_{i+1}, \ldots, w_n)$, where $i$ is a position for separating a sentence to $S_1$ and $S_2$. If $S_1$ and $S_2$ is independent. The function of joint probability in Eq. 1 can be written as:

$$p(w_1^n) = p(w_1^i)p(w_{i+1}^n). \tag{3}$$

For considering the situation when $S_1$ and $S_2$ are not independent, we give the function as follows:

$$p(w_1^n) = \sum_{j=1}^{K} p(y_j)p(w_1^i|y_j)p(w_{i+1}^n|y_j) \tag{4}$$

where $y_j$ is the conditional variable, $y_i \in \mathbb{Y}$, and $\mathbb{Y} = \{y_1, \ldots, y_K\}$. When $K$ is equal to 1, Eq. 4 can be reduced to Eq. 3, since $p(y_1) = 1$.

In language modeling, if $K$ is equal to 1, then $p(w_1^n) = p(w_1^i)p(w_{i+1}^n)$, which means that there is no dependency between two context pieces $S_1$ and $S_2$. The higher $K$ is, the further two context pieces ($S_1$ and $S_2$) are from being separable, meaning that $S_1$ and $S_2$ has stronger dependency.

### 3.2 SEPARATION RANK IN TSLM

In this section, we define the separation rank based on $K$ in TSLM. First, we consider $p(w_1^n)$ in Eq. 2 as a probability function. Then, the separation rank is defined on the factorization of $p(w_1^n)$. Such a factorization is carried on the matricization of weight tensor $\mathcal{T}$ in Eq. 2. The matricization process is to put the entries from a tensor into a matrix. It turns out that the factorization of $p(w_1^n)$ is reduced to the factorization of of the matrix form of the tensor $\mathcal{T}$, as shown in the following claim.

**Claim 1** *The factorization of $p(w_1^n)$ in Eq. 2 can be obtained by the Singular Value Decomposition (SVD) on the matrix $[\![T]\!]_{(S_1,S_2)}$. After the decomposition, the matrix $[\![T]\!]_{(S_1,S_2)}$ is written as follows:*

$$[\![T]\!]_{(S_1,S_2)} = \sum_{j=1}^{K} \lambda_j \boldsymbol{v}_j \boldsymbol{u}_j^T \tag{5}$$

*where $K$ is number of non-zero singular values.*

**Proof 1** *Our proof can be found in Supplementary Appendices A.1 submitted.*

In Eq. 4, each different position $i$ corresponds to different partitions $(S_1, S_2)$ and consequent matricizations of tensor $T$. Thus there could be different decomposition and different numbers of non-zero singular values. In order to find a measurement for contextual dependency, we define the separation rank as the minimum value of $K$ in TSLM. The claim is given as follows.

**Claim 2** *Suppose a sentence $S=(w_1,\ldots,w_n)$ can be split into two disjoint subset $S_1=(w_1,\ldots,w_i)$ and $S_2=(w_{i+1},\ldots w_n)$, where $i \in \{1,\ldots,n-1\}$ is any position. The separation rank of function in Eq. 2 in tensor space language model is formulated as:*

$$sep_{(S_1,S_2)}(p(w_1^n)) = min\{K \in \mathbb{N} \mid p(w_1^n) = \sum_{j=1}^{K} p(y_j)p(w_1^i|y_j)p(w_{i+1}^n|y_j)\} \tag{6}$$

**Proof 2** *Our detailed proof can be found in Supplementary Appendices A.2 submitted.*

In general, the separation rank in TSLM $sep_{(S_1,S_2)}(p(w_1^n))$[1] have been defined in Eq. 6.

## 4 LOWER BOUND ANALYSIS OF SEPARATION RANK FOR RECURRENT NEURAL NETWORK STRUCTURE

In this section, by providing the lower bound of the separation rank in TSLM, we investigate the interpretable mechanism in recurrent neural network language modeling. Specifically, we analyze the relation between the separation rank (reflecting the contextual complexity) and the network architecture (e.g., network depth and width). Recall that tensor space language model (TSLM) can derive a recurrent language modeling process from via the tensor decomposition of high-order tensors (Zhang et al., 2019). Based on the above analysis, we will establish the relation between separation rank in TSLM (see Sec.3.2) and parameters of recurrent neural language models.

Different partition position $i$ in Eq. 4 corresponds to different partition $(S_1, S_2)$ in Eq. 6. If the partitions $(S_1, S_2)$ are different, the values of $K$ will be different. Without loss of generality, we set an equal partition, i.e., $i = \lfloor \frac{n}{2} \rfloor$. When a sentence is partitioned equally, $K$ value in Eq. 6 is larger than the actual separation rank since it is the minimal $K$.

Then, based on the equal partition criteria, step by step, we propose our theoretical results regarding the lower bound of the separation rank. Such a lower bound is expected to reflect the relation between the ability for modeling contextual dependencies and the model structure (e.g., network depth and width) in the language modeling process. Specifically, if a sentence is modeled by a recurrent network architecture, the lower bound of the separation rank in TSLM can be computed by some parameters (i.e., hidden units, the number of layers and the number of words in a sentence) in recurrent network architecture. In the following, we first show such quantitative result when the number of layers is one (i.e., $L = 1$), and then we provide our findings when $L > 1$.

**Claim 3** *In tensor space language model (TSLM), when the number of layer is one (i.e., $L$=1,) the separation rank in TSLM can be given as follows:*

$$sep_{(S_1,S_2;i=\lfloor \frac{n}{2} \rfloor)}(p(w_1^n)) \geq R, \;\; for \;\; L = 1. \tag{7}$$

*where $R$ is the number of hidden units, $(S_1, S_2)$ is the case of equal partition.*

**Proof 3** *The detailed proof can be found in Supplementary Appendices A.3 submitted.*

---

[1]The separation rank can not be accuracy computed in experiments.

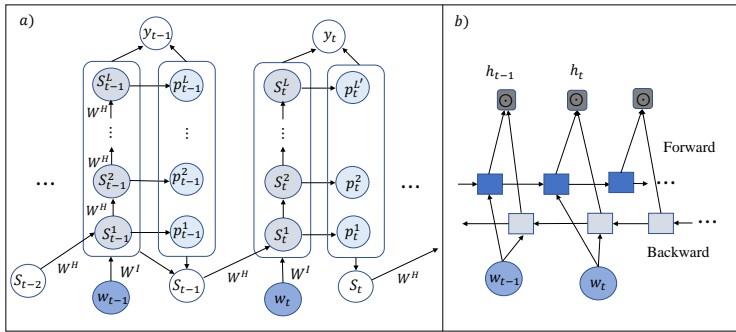

Figure 1: The diagram about a bidirectional adaptive RNN. $a)$ is a single direction adaptive RNN, $b)$ is a schematic diagram of bidirectional RNN.

In order to show the influence of network depth, we need to further extend the tensor space language model using recurrent network with deep layers, i.e., $L > 1$. We derive the lower bound, i.e., the separation rank in TSLM in case of the $L \geq 2$. Further details will be expound in Supplementary Appendices A.4. The lower bound will be shown as follows:

$$sep_{(S_1, S_2; i = \lfloor \frac{n}{2} \rfloor))}(p(w_1^n)) \geq \min\{\frac{(R + F - 1)!}{F!(R - 1)!}, m^{\lfloor \frac{n}{2} \rfloor}\} \tag{8}$$

where $F = \frac{(\lfloor n/2 \rfloor + L - 2)!}{(L-1)!(\lfloor n/2 \rfloor)!}$, $R$ is the number of the hidden units, $n$ is the number of words in a sentence, $m$ is the dimension of a word vectors, $L$ is the number of layers and $(.)!$ is the symbol of factorial. $(S_1, S_2; i = \lfloor \frac{n}{2} \rfloor)$ is the case of equal partition.

Next, we explore the relation between the lower bound and network parameters in detail. From Eq. 8, we can find that increasing both $R$ and $L$ can all improve the lower bound of separation rank in TSLM, i.e., the separation rank of a recurrent network language model will be increased. In order to illustrate this quantitative relationship between separation rank in TSLM and the recurrent network architecture (i.e., $R$ and $L$). In Eq. 8, we see that if $L$ was increased, it is observed that the lower bound will be increased by an exponential level. Through increasing $R$, the lower bound is increased by an linear level. Table 2 in supplementary appendices shows the numerical relationship.

## 5 BIDIRECTIONAL ADAPTIVE RECURRENT NEURAL NETWORK

The effectiveness of neural network depth is an important research problem (Vinayakumar et al., 2017). In Sec.4, the analysis of lower bound has described that increasing the number of layer can be more effective to improve the ability of modeling contextual dependency. In the processing for modeling a sequence of words, it is important to search an adaptive network depth. Under most circumstances, researches choose a fixed number of network layers $L$ to model text. In this section, we describe how to construct an adaptive bidirectional recurrent neural network.

### 5.1 STRUCTURE CONSTRUCTION

As in Eq. 6, we use a backward recurrent neural network language model as the right function $p(w_{i+1}^n | y_j)$. After that, a forward recurrent neural network network language model $\mathcal{R}_f$ can be seen as the left function $p(w_1^i | y_j)$. In experiment, we first train the backward function and get the hidden output. When it predicts the next words in forward language model, the hidden vectors from the backward process are used to multiply the hidden vectors from the forward process. In Figure 1, we provide the diagram for the bidirectional adaptive RNN.

In Figure 1 a), $\boldsymbol{w}_{t-1}$ and $\boldsymbol{w}_t$ are the inputs of word vectors. $\boldsymbol{S}_{t-2}$, $\boldsymbol{S}_{t-1}$ and $\boldsymbol{S}_t$ are some vectors from the outputs of hidden layer. Eq. 9 shows how to compute the $S_t$.

$$\boldsymbol{S}_t = \sum_{i=1}^{L'} p_t^i \boldsymbol{S}_t^i \tag{9}$$

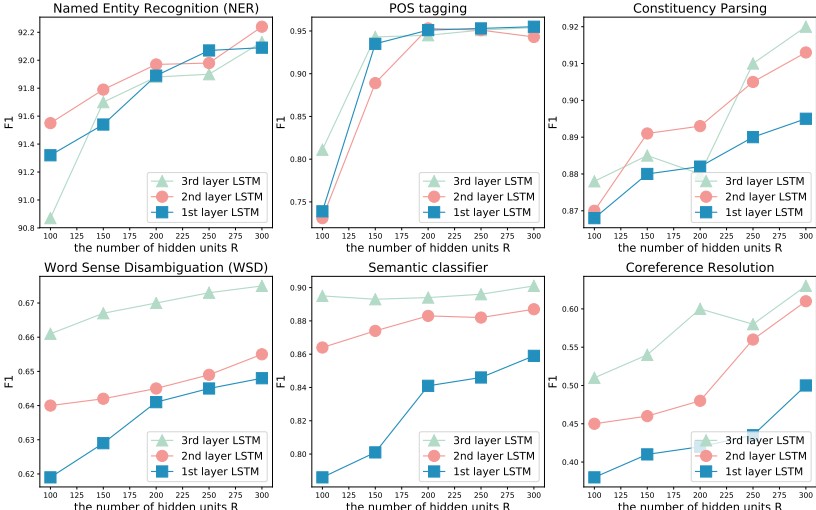

Figure 2: Results on the short-range (first row) and long-range (second row) dependencies tasks. The displayed results show that the increasing of $R$ is more helpful in short-range dependencies tasks (NER, Constituency Parsing and POS tagging), while the increasing of $L$ is more effective for long-range dependencies tasks (WSD, Semantic classifier and Coreference Resolution), compared with the increasing of $R$.

where $p_t^i$ is a scalar value and presents the probability value of $\boldsymbol{S}_t^i$. The probability value can be computed as: $p_t^i = \boldsymbol{v}_t^T \boldsymbol{S}_t^i$, where $\boldsymbol{v}_t$ is the trainable vector and can be initialized at the start of training. $p_t^i$ is equal to the inner product of $\boldsymbol{v}_t$ and $S_t^i$.

## 5.2 DYNAMIC HALTING

Since we wish to explore a recurrent neural network with the minimum computational cost. Here, we first give out some basic assumptions. Since Eq. 16 is to get the minimal $K$, and increasing the depth $L$ can improve the ability of modeling contextual dependency by an exponential level in Eq. 8, we can assume the computing of the depth $L$ have a straight relation. Inspired by Adaptive Computation Time to dynamically halt changes in certain representations (Graves, 2016), we design the condition of halting as follows. There are two kinds of halting conditional in our model. First, the times of computing is raised to the maximum number of calculations. Second, the steps of computing is satisfied as $N(t)=min\{l : \sum_i^l p_t^i \le 1 - \epsilon\}^2$.

## 6 EXPERIMENTS

The about analysis has shown that improving $R$ and $L$ can improve the lower bound at different levels. The lower bound can imply the ability of recurrent neural network language models for modeling the contextual dependencies. For different NLP tasks, we need to choose the network structure with different modeling capabilities. We carry out six classification tasks and an intra-sentence similarity experiment to verify our theoretical, respectively.

## 6.1 NATURAL LANGUAGE PROCESSING TASKS

This set of experiments involves six tasks including Named Entity Recognition (NER), POS tagging, Constituency parsing, Word Sense Disambiguation (WSD), Sentiment analysis and Coreference Resolution. According to previous study (Peters et al., 2018a), NER, constituency parsing and POS tagging are more required to model short-range dependency, while WSD, Semantic classifier and Coreference Resolution are required to model long-range dependency.

---

[2]$\epsilon$ is generally taken as $0.001$ in experiments.

Table 1: Results on sentences classification dataset. Accuracy (acc) is used as the evaluation criterion. * represents the results achieved by ourself.

| Model | MR | SST2 | CR | SUBJ | MPQA |
|---|---|---|---|---|---|
| FastText (Joulin et al., 2016) | 0.765 | 0.788 | 0.789 | 0.916 | 0.874 |
| Sent2Vec (Gupta et al., 2019) | 0.763 | 0.802 | 0.791 | 0.912 | 0.872 |
| Transformer (Vaswani et al., 2017) | 0.731* | 0.761* | 0.762* | 0.863* | 0.723* |
| BiLSTM (Conneau et al., 2017) | 0.775 | 0.807 | 0.813 | 0.896 | 0.887 |
| Adaptive-LSTM | 0.784* | 0.816* | **0.821*** | **0.92*** | 0.898* |
| Adaptive-BiLSTM | **0.790*** | **0.818*** | 0.820* | 0.917* | **0.904*** |

Based on our analysis in the previous section, we have two hypotheses to be tested. (1) For tasks with short-range dependencies required, improving $R$ can to some extent meet the modeling requirement, while improving the network depth $L$ could be not necessary; (2) For tasks requiring long-range dependencies, it is necessary to improve $L$ to improve the ability of modeling dependencies.

These tasks be considered as classification tasks, for which the experimental results are described in Sec. 6.1.2. In language modeling process, LSTM is a commonly used recurrent network structure. Therefore, we mainly carried out experiments based on the structure of LSTM and its variants.

### 6.1.1 TESTING TASKS

**POS tagging** We conduct experiments on the Wall Street Journal of the Penn Treebank dataset (PTB) (Marcus et al., 1993). The model Ling et al. (2015) is used, which using bidirectional LSTMs.
**Named Entity Recognition** The CoNLL 2003 NER task (Tjong Kim Sang & De Meulder, 2003) consists of newswire from the Reuters RCV1 corpus tagged with four different entity types. We follow the language model augmented sequence taggers (TagLM) (Peters et al., 2017).
**Constituency parsing** The Penn Treebank (Marcus et al., 1993) which contains phrase structure annotation is used in this application. We use the model named Reconciled Span Parser, which used the biLM representation.
**Word Sense Disambiguation** We follow this model (Kågebäck & Salomonsson, 2016) which used the bidirectional LSTMs architecture to compute a probability distribution over the possible senses corresponding to that word. The dataset of Senserval-2 (Edmonds & Cotton, 2001) is used.
**Coreference Resolution** An end-to-end coreference resolution model (Lee et al., 2017) has been proposed. We conduct experiments on the OntoNotes Release 5.0 benchmark (Pradhan et al., 2012).
**Sentiment analysis** We use the IMDB dataset (Maas et al., 2011) with two categories. We perform classification using a standard bidirectional LSTM with different hidden units and layers.
The experimental setup of the six testing tasks can be found in Supplementary Appendices A.5.1.

### 6.1.2 EXPERIMENTAL RESULTS AND ANALYSIS

The results on six NLP tasks are reported in Figure 2. The results of short-range dependency and long-range dependency tasks are shown as the first row and second row, respectively.

In short-range dependency tasks, using 1-layer LSTM, its $F_1$ scores increase along the increasing of the number of hidden units $R$. There are similar experimental phenomenons when using 2-layer and 3-layer LSTMs. For each $R$, after $L$ be increased, we observe that increasing $F_1$ score is not very obvious. These phenomena mean that for the increasing of $F_1$, the increasing of $R$ is more obvious than the increasing of $L$. Therefore, the results in the first row of Figure 2 verify our first hypothesis about short-range dependent tasks.

In the second row of Figure 2, these trends of line charts show that the increasing for the number of hidden units $R$ and the number of layers $L$ can help to increase the $F_1$ score, while the effects of increasing $L$ is more obvious, for long-range dependency tasks. The $F_1$ scores are the highest through using 3-layer LSTMs in long-range dependency tasks. In order to investigate different effects of increasing $R$ and $L$ on $F_1$ scores, we have detailed observations: When $L$ is the largest (saying 3), the increasing of $F_1$ score is getting relatively slow along with the increasing of $R$, as shown in the results of semantic classifier (the middle in the second row in (2)). These phenomena and trends verify the second hypothesis. It is that increasing $L$ can improve the network modeling

ability more effectively than increasing $R$ for long-range dependency tasks, In summary, $R$ is more necessary to be increased for the short-range dependency tasks, while $L$ is relatively more necessary to be increased in the long-range dependency tasks. Such experimental results and analysis are consistent with our theoretical analysis in Sec 4 and testing hypotheses in this section.

## 6.2 INTRA-SENTENCE SIMILARITY TASK

In this section, we present a detailed empirical study about the effect of different network architecture (i.e., network width and the number of layers) on the representation ability of contextual word embeddings. Based on previous analysis in Sec. 4, our hypothese are as follows. (1) Increasing $R$ can improve the similarity between short-range words. (2) Increasing $L$ can improve the similarity between long-range words, (3)Increasing $L$ is more effective for modeling the similarity between words than the increasing of $R$.

1-Billion Word Benchmark dataset Chelba et al. (2013) is used, and it has approximately $800M$ tokens of news crawl data from WMT 2011. Different recurrent network architectures (i.e., using different layers and the number of hidden units) are trained in $EMLo$, which uses bidirectional LSTM structure. The training of $EMLo$ model will be stopped when the average perplexity is $40$ in test dataset. After that, we select a sentence from the test data and compute the intra-sentence similarity. We use cosine similarity, which is a popular way to measure the similarity between word vectors encoded in different layers of $EMLo$. Figure 3 in Supplementary Appendices shows the intra-sentence contextual similarity among words in a sentence. The details of the experimental results and analysis will be discussed in Supplementary Appendices A.5.2.

## 6.3 BIDIRECTIONAL ADAPTIVE RECURRENT NEURAL NETWORK

In this section, we test the bidirectional adaptive recurrent neural network on five text classification tasks. Text classification task is to refer to the process of automatically determining the text category based on the textual content under a given classification system. The Precision is used to evaluate the quality of the model. We test our model on MR, SST-2, CR, MPQA and TREC, respectively. The experiment results of this part is shown in Table 1.

Table 1 shows the results about the adaptive network in sentence classification tasks. Results shows that the Adaptive-BiLSTM model achieves better accuracy (acc) in these datasets than BiLSTM. Our method achieves a larger improvement on the MPQA dataset. The adaptive bi-LSTM (bidirectional LSTM) and the model of adaptive LSTM achieve very close results.

## 7 CONCLUSION AND FURTHER WORK

In this paper, we provide quantitative analysis for modeling contextual dependencies in recurrent language model and its variants. First, we define the separation rank in tensor space language model to measure the contextual dependencies of a sentence. Then, we analyze the lower bound of the separation rank in the recurrent neural network based LM. Based on the lower-bound analysis, we explain the quantitative relation between the value of separation rank and the network structure. For certain NLP tasks requiring different ranges of contextual dependencies, the lower-bound analysis will suggest different settings for the network depth and width. These suggestions based on the quantitative lower-bound analysis, are verified by various NLP tasks. In order to find the number of layers with both minimum cost and great performance results, we designed an adaptive RNN and verify the effectiveness of the method on text classification tasks.

In the future work, we can design feasible methods to calculate the contextual dependencies, and use such signals to design automatic language modeling approaches, which can be adaptively controlled for specific NLP tasks with certain requirement for the contextual dependencies. In addition, we will explore the use of tensor space to explain or advance the more recent language modeling approach, e.g., Transformer and BERT.

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

# A  SUPPLEMENTARY APPENDICES

## A.1  CLAIM 1

The factorization of $p(w_1^n)$ in Eq.2 (in paper) can be obtained by the Singular Value Decomposition (SVD) on the matrix $[\![T]\!]_{(S_1,S_2)}$. After the decomposition, the matrix $[\![T]\!]_{(S_1,S_2)}$ is written as follows:

$$[\![T]\!]_{(S_1,S_2)} = \sum_{j=1}^{K} \lambda_j \boldsymbol{v}_j \boldsymbol{u}_j^T \tag{10}$$

where $K$ is the number of non-zero singular values.

*Proof sketch of this claim*

In the tensor space language model, the input tensor is a rank-one tensor. To make it easy to understand, we assume that the word vector is one hot vector. The high order tensor $A$ can be given by the operator of tensor product. Therefore, the tensor $A$ is one hot tensor, which means that the tensor only have one element is 1 and other elements are zero. The basic form of tensor space language can be re-written an element $T_{d_1,\dots,d_n}$ from the tensor $T$. After that, the element can be consider a value in a matrix which can be given by the matricization of the tensor $T$. For all elements of the matrix, we can consider a method of SVD to get the value of $K$.

It would be helpful to understand if the definition of matricization was given. Therefore, There is the definition of matricization as follows.

**Definition 1** *Suppose $T \in \mathbb{R}^{m \times \dots \times m}$ is a tensor of order $n$, and let $(S_1, S_2)$ be a paratition of a set $\{1, \dots, n\}$, and $S_1$ and $S_2$ are disjoint subsets. The matricization of $T$ denoted as $[\![T]\!]_{S_1,S_2}$, is the $m^{|S_1|}$-by-$m^{|S_2|}$ matrix holding the entries of $T_{d_1,\dots,d_n}$ is placed in row index $1 + \sum_{t=1}^{|S_1|}(d_{it} - 1)m^{|S_2|-t}$ and column index $1 + \sum_{t=1}^{|S_2|}(d_{j_t} - 1)m^{|S_2|-t}$, where $i \in S_1$ and $j \in S_2$.*

**Proof 4** *. In subsection 2.2, Eq.2 is the basic form of tensor space language model. In Eq.2, $A$ is a rank-one tensor $A = \boldsymbol{w}_1 \otimes \dots \otimes \boldsymbol{w}_n$. We set each word as an one-hot vector. The tensor $A$ is a tensor with only one entry being one and the other entries being zero. The Eq.2 can be rewritten as:*

$$p(w_1^n) = \sum_{d_1,\dots,d_n=1}^{|V|} T_{d_1\dots d_n} A_{d_1\dots d_n} \tag{11}$$

*where*

$$A_{d_1\dots d_2} = \begin{cases} 1, & d_k = index(w_k, V), \forall k \in [n] \\ 0, & otherwise \end{cases} \tag{12}$$

*which means tensor $A$ is an one-hot tensor, and $index(w, V)$ means the index of $w$ in vocabulary $V$. $k \in [n]$ also is denoted as $k \in \{1, \dots, n\}$. The probability of a sentence $s$ is:*

$$p(w_1^n) = \sum_{d_1,\dots,d_n=1}^{|V|} T_{d_1\dots d_n} A_{d_1\dots d_n} \tag{13}$$
$$= T_{d_1,\dots,d_n}, \ d_k = index(w_k, V), \forall k \in [n]$$

*Therefore, the factorization of $p(w_1^n)$ in Eq. 2(in paper) can be achieved by two steps as follows.*

*Firstly, the matricization of weight tensor $T$ can be achieved by the matricization definition 1 and represented as $[\![T]\!]_{(S_1,S_2)}$. Secondly, we can perform singular value decomposition (SVD) on matrix $[\![T]\!]_{(S_1,S_2)}$.*

$$[\![T]\!]_{(S_1,S_2)} = \boldsymbol{U}\boldsymbol{S}\boldsymbol{V}^T \tag{14}$$

*Therefore, we can compute each element in the matrix $[\![T]\!]_{(S_1,S_2)}$, in other words, we can get every entry from tensor $T$ and represent as:*

$$T_{(d_1\dots d_i),(d_{i+1}\dots d_n)} = \sum_{v=1}^{K} \lambda_v U_{v,d_1,\dots,d_i} V_{v,d_{d_{i+1},\dots,d_n}} \tag{15}$$

where $U_{v,d_1,\ldots,d_i}$ is the element from the matrix $U$ and the $U_{v,d_1,\ldots,d_i}$ is the element from the matrix $V$. Let $V_{v,d_1,\ldots,d_i}$ and $U_{v,d_1,\ldots,d_i}$ can be given by $p(w_1^i|y_v)$ and $p(w_{i+1}^n|y_v)$, respectively. $\lambda_v$ is the singular value and is seed as $p(y_v)$. We can represent Eq. 15 as Eq. 4 (in paper).

## A.2  CLAIM 2

Suppose a sentence $S = (w_1, \ldots, w_n)$ can be split into two disjoint subset $S_1 = (w_1, \ldots, w_i)$ and $S_2 = (w_{i+1}, \ldots w_n)$, the separation rank of function Eq. 2 by tensor space language model is formulated as following:

$$sep_{(S_1,S_2)}(p(w_1^n)) = min\{K \in \mathbb{N} \mid p(w_1^n) = \sum_{j=1}^{K} p(y_j)p(w_1^i|y_j)p(w_{i+1}^n|y_j)\} \qquad (16)$$

where $K$ is the number of no-zero singular values, $i$ is the position of separation.

Before proving the claim, we should give out a definition as reported in (Cohen & Shashua, 2016b).

**Definition 2** *Let $(I_1, I_2)$ be a partition of ordered input indexes, i.e. $I_1$ and $I_2$ are disjoint subsets of $[N]$ whose is the set of all indexes. $I_1 = i_1, \ldots, i_{|I_1|}$ where $i_1 < \ldots < i_{|I|}$, and similarly $I_2 = j_1, \ldots, j_{|I_2|}$ where $j_1 < \ldots < j_{|J|}$. For a function $h : (\mathbb{R}^s)^N \to \mathbb{R}$, the separation rank w.r.t the partition $(I_1, I_2)$ is defined as follows:*

$$sep(h; I_1, I_2) := min\{R \in \mathbb{N} \cup \{0\} : \exists g_1 \ldots g_R, g_1' \ldots g_R' \ s.t. \ h(x_1, \ldots, x_N) =$$

$$\sum_{v=1}^{R} g_v(x_{i_1}, \ldots, x_{i_{|I_1|}})g_v'(x_{i_1}, \ldots, x_{i_{I_2}})\}$$

*It is the minimal number of summands that together give h, where each summand is separable and equal to a product of two function. $g_i$ and $g_i'$ are the functions, $g_i:(\mathbb{R}^s)^{|I_1|}$ and $g_i':(\mathbb{R}^s)^{|I_2|}$.*

*Proof sketch of this claim*

According to Eq. 15 and Definition 2, it will be easy to prove the result that the separation rank between two different inputs $S_1$ and $S_2$ is equal to the minimum value of $K$ for different separation.

**Proof 5** *In Eq. 15, if we set the $\sqrt{\lambda_v}U_{v,d_1,\ldots,d_i}$ is equal to $g_v(\boldsymbol{w}_1, \ldots, \boldsymbol{w}_i)$ and $\sqrt{\lambda_v}V_{v,d_1,\ldots,d_i}$ is equal to $g_v'(\boldsymbol{w}_{i+1}, \ldots, \boldsymbol{w}_n)$. According to the definition 2, we can give the function of the separation rank in tensor space language model.*

$$sep_{(S_1,S_2)}(p(w_1^n)) = min\{R \in \mathbb{N} : \exists g_1 \ldots g_R, g_1' \ldots g_R' \ s.t. \ p(w_1^n) =$$

$$\sum_{v=1}^{R} g_v(w_1, \ldots, w_i)g_v'(w_{i+1,\ldots,w_n})\} \qquad (17)$$

*In a word, it is the minimal number of summands that together give $p(w_1^n)$, where each summand is separable and is equaltion to a product. After that, for different $(S_1, S_2)$ on a sentence, $K$ is different. Therefore, we can define the minimum value of $K$ as the separation rank in tensor space.*

## A.3  CLAIM 3

In tensor space language model (TSLM), when the number of layer is one (i.e., $L$=1,) the separation rank in TSLM can be given as follows:

$$sep_{(S_1,S_2;i=\lfloor \frac{n}{2} \rfloor)}(p(w_1^n)) \geq R, \ \ for \ \ L = 1. \qquad (18)$$

where $R$ is the number of hidden units, $(S_1, S_2)$ is the case of equal partition, i.e., the position $i$ in Eq.4(in paper) is equal to $\lfloor \frac{n}{2} \rfloor$.

*Proof sketch of this claim*

We can consider the RNN is a function $y$. In addition, the function can be re-written as a new form which is the sum of $R$ groups of factor functions product. $R$ is the number of hidden units, and also the separation rank in RNNLM. It meets the definition of separation rank in definition 2.

**Proof 6** *The recurrent network language model of the number of layers L=1 have been given:*

$$\begin{aligned} \boldsymbol{h}_t &= W\boldsymbol{h}_{t-1} \odot U\boldsymbol{w}_t \\ y_t &= V\boldsymbol{h}_t \end{aligned} \tag{19}$$

*where $\boldsymbol{w}$ is the current word inputs, $\boldsymbol{h}_t$ is the current hidden state, $\boldsymbol{w}_t$ is the word vector of $t$-th word. $y_t$ is probability distribution of the current state (i.e., the joint probability distribution). The $\odot$ denotes the element-wise multiplication between vector (i.e., Hadamard product). $W$, $U$ are square matrix in TSLM and the dimension of $W$ is the number of hidden units $R$. We set the $W = I$. $V$ is the full connection matrix. Further, we can compute current hidden state as follows:*

$$\begin{aligned} \boldsymbol{h}_t &= U\boldsymbol{w}_t \odot W\boldsymbol{h}_{t-1} \\ &= U\boldsymbol{w}_t \odot W(U\boldsymbol{w}_{t-1} \odot \boldsymbol{h}_{t-2}) \\ &\Rightarrow U\boldsymbol{w}_t \odot U\boldsymbol{w}_{t-1} \odot h_{t-2} \ \ (W = I) \\ &\dots \\ &\Rightarrow U\boldsymbol{w_t} \odot \dots \odot U\boldsymbol{w}_{t-\frac{t}{2}} \odot \boldsymbol{h}_{t/2} \end{aligned} \tag{20}$$

*where we set the $\boldsymbol{h}_{\frac{t}{2}}$ is the function $g(\boldsymbol{w}_1, \dots, \boldsymbol{w}_{t/2})$ and the $U\boldsymbol{w_t} \odot \dots \odot U\boldsymbol{w}_{t-\frac{t}{2}}$ is the function $g'(w_{\frac{t}{2}+1}, \dots, \boldsymbol{w}_t)$. After that, we can compute each probability in the probability distribution $y_t$. The process is as follows:*

$$\begin{aligned} y_t &= V\boldsymbol{h}_t \\ y_t(i) &= \sum_{j=1}^{R} V_{i,j} g_j(\boldsymbol{w}_1^{t/2}) g'_j(\boldsymbol{w}_{t/2+1}^n) \end{aligned} \tag{21}$$

*where $i \in 1, \dots, |V|$ is the index in the vector $y_t$ and $|V|$ is the size of vocabulary. $V$ is a matrix of $R$ by $|V|$. At the same time, $i$ means also the position of the last word of a sentence in the vocabulary. Therefore, the $p(w_1^t)$ can be written as:*

$$p(w_1^t) = y_t(i)$$

*Let $n$ be equal to $t$, the inputs can be written $(\boldsymbol{w}_1, \dots, \boldsymbol{w}_{\lfloor n/2 \rfloor}) = S_1$ and $(\boldsymbol{w}_{\lfloor n/2 \rfloor+1}, \dots, \boldsymbol{w}_n) = S_2$. According to the definition of the separation separation rank in the definition 2, we can get the result as follow:*

$$sep_{(S_1, S_2)}(p(w_1^n)) \geq R \tag{22}$$

*Therefore, the separation rank in TSLM is $R$ when we using the 1-layer recurrent network.*

## A.4 SEPARATION RANK IN TENSOR NETWORK

Let $y^L$ be the function computing the output after $t$ time-steps of an recurrent tensor network with $L$ layers, $R$ hidden layer channel numbers, weights denoted by tensor $\mathcal{T}$ and initial hidden states $h^{0,l}, l \in [L]$. The relations between start-end separation rank and the TN architecture (Levine et al., 2017) are shown as follow:

$$\begin{aligned} seq_{(T_1,T_2)}(y^1) &= min\{R, m^{T/2}\}, && L = 1 \\ seq_{(T_1,T_2)}(y^2) &\geq \left( \binom{min\{m, R\}}{T/2} \right), && L = 2 \end{aligned}$$

where $\left( \binom{min\{m,R\}}{\lfloor T/2 \rfloor} \right)$ is the multiset coefficient, given in the binomial form by $\binom{min\{m,R\}+\lfloor T/2 \rfloor-1}{\lfloor T/2 \rfloor}$. $R$ is the hidden channel numbers, $T$ is the length of sequence inputs, and $m$ is the dimensions of each model in high-order tensor. $(T_1, T_2)$ is an average separation of time steps $T$. For clearly understand the multi-set coefficient, We can understand that select $\lfloor \frac{T}{2} \rfloor$ types from $R' = min\{R, m\}$ class elements. Now, a lower bound of $\left( \binom{R}{\lfloor T/2 \rfloor} \right)$ on the separation rank of depth $L = 2$ have been provided. In the following, The conjecture that a tighter lower bound

holds for networks of depth $L > 2$, the form of which implies that the dependency capacity of deep recurrent networks grows combinatorially with the network depth:

$$seq(y^L) \geq min\{\left(\binom{R}{\left(\binom{\lfloor T/2 \rfloor}{L-1}\right)}\right), m^{\lfloor T/2 \rfloor}\} \ \ L \geq 2 \tag{23}$$

where $R$ is the number of hidden units, $L$ is the number of layers. $n$ is the number of words in a sentence. In language model, for easy to analysis the relation between separation rank and parameter from the recursive networks architecture. We set the $R = m$. We do not consider time series. Let's set the $T$ in Eq. 23 to $n$. $n$ is the numbers of words in a sentence. Considering the constraints of tensor dimensions, the maximum of TSLM separation rank is $m^{\lfloor T/2 \rfloor}$. $T$ is the temple in RNN, we suppose that it is the equal $n$, $n$ is the length of a sentence. In the case where $R$ and $L$ are controllable, the value to the left of the minimum function 23 is less than the value to the right. After that, we can get the relations as follow.

$$sep_{(S_1, S_2; i=\lfloor \frac{n}{2} \rfloor))}(p(w_1^n)) \geq \min\{\frac{(R+F-1)!}{F!(R-1)!}, m^{\lfloor \frac{n}{2} \rfloor}\}$$

$$where \ F = \frac{(\lfloor n/2 \rfloor + L - 2)!}{(L-1)!(\lfloor n/2 \rfloor)!} \tag{24}$$

where the $R$ is the number of hidden size, $n$ is the number of words in a sentence, and $L$ is the number of layers. Note that ! is the symbol of factorial.

## A.5 EXPERIMENT

Table 2: The quantitative relation between the separation rank in TSLM and the parameters (i.e., $R$ and $L$). Increasing $L$ and $R$ approximate the separation rank of recurrent network exponentially and linearly, respectively. Example: $n = 4$ (i.e., sentence length), $R \in \{14, 15, 16, 17\}$ and $L \in \{1, 2, 3, 4\}$.

| $R$ | 14 | 15 | 16 | 17 |
|---|---|---|---|---|
| $L = 1$ | 14 | 15 | 16 | 17 |
| $L = 2$ | 455 | 560 | 680 | 816 |
| $L = 3$ | 8008 | 12376 | 18564 | 27132 |
| $L = 4$ | $1.9*10^4$ | $4.3*10^4$ | $9.2*10^4$ | $3.5*10^5$ |

### A.5.1 EXPERIMENTAL SETUP

Sec. 4 analyzed the relation between contextual dependencies and recurrent network architecture through $L$ and $R$. Specifically, the lower bound of the recurrent language model modeling the sentence contextual dependencies will be increased by increasing $L$ ($L \in \{1, 2, 3\}$) and $R$ ($R \in \{100, 125, 150, 175, 200, 225, 250, 300\}$) in long-range/short-range dependency tasks. Recall that our hypotheses are: For short-range dependent tasks, increasing $R$ is sufficient to reflect the network modeling ability, while for long-range dependent tasks, increasing $L$ can improve the network modeling ability more effectually than increasing $R$. We test our hypotheses through the trend of $F_1$ score. $F_1$ is a mainly and common indicator of evaluation in these task.

### A.5.2 ANALYSIS FOR INTRA-SENTENCE SIMILARITY TASK

Our experiments focus on the changing trends of the similarity between word embeddings along with the increasing of $R$ and $L$. This experiment is different from the experiments reported by Peters et al. (2018a). They only consider the difference between upper layer and lower layer and do not consider the changing of network depth $R$.

Figure 3 shows the intra-sentence contextual similarity between all words in a sentence. In Figure 3, we obtain several observations. When the number of hidden units $R$ is fixed, we observe that the bright area is larger along with the increase of the number of layers $L$ from 1 to 3. The bright area indicates the contextual dependency between words in our work. Relatively speaking, the larger the

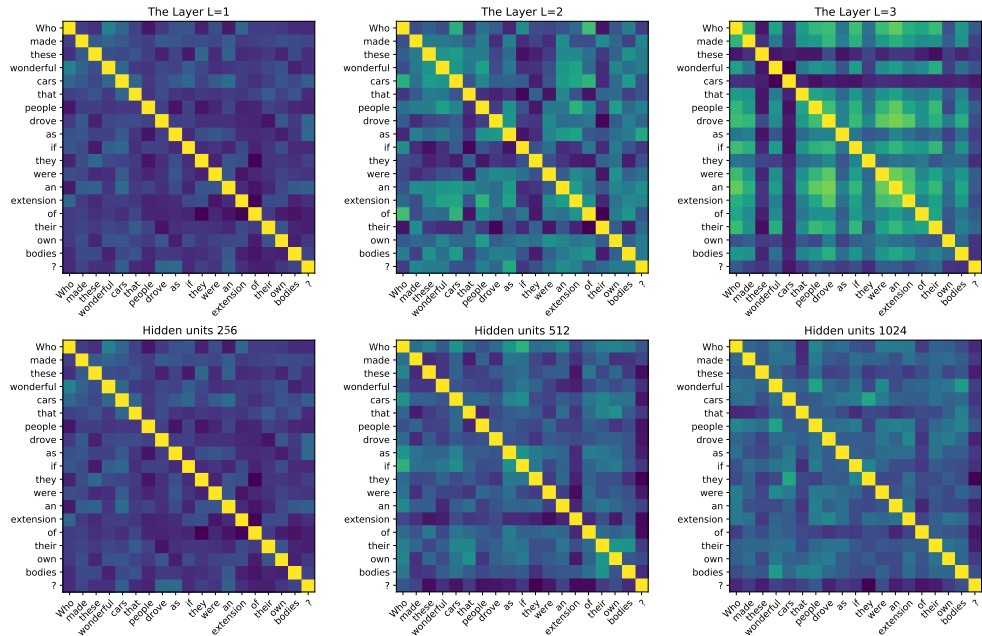

Figure 3: Visualization of contextual similarity between all words pairs in a single sentence. For the first row, we set the hidden layer units to be 256, with layers from 1 to 3. For the second row, we set the number of layers to 1, and the hidden layer units (256, 512 and 1024, respectively). The lighter yellow-colored areas indicate the higher contextual similarity.

Table 3: The statistical results about the average of the context similarity. Layers and hidden units are represented by $L$ and $R$ respectively.

| $R(for\ L = 1)$ | 256 | 512 | 1024 |
|---|---|---|---|
| average similarity | 0.25 | 0.36 | 0.38 |
| $L(for\ R = 256)$ | 1 | 2 | 3 |
| average similarity | 0.25 | 0.39 | 0.48 |

bright region is, the stronger the ability of a language model modeling the contextual dependency. When the number of layers is fixed, we observe that the bright area is increased along with the increasing of the number of hidden units from 256 to 512. It may mean that short-range contextual dependency can be captured by adding the number of hidden units. These experimental phenomena verify the first hypothesis and the second hypothesis.

Correspondingly, we can also observe from Table 3 that the larger the bright area is, the larger the average of contextual similarity is. In Table 3, when the $L$ is equal to 3 (when $R$=256), the average of contextual similarity is the largest. Combined Figure 3 and Table 3, we can find that the increasing of the number of layers can more effective to capture long-range similarity information. The results in Table 3 and in Figure 3 verify the third hypothesis.

