# OpenReview forum: "Interpretable Network Structure for Modeling Contextual Dependency"
_ICLR.cc/2020/Conference — Reject_

### Official Review · AnonReviewer1 · 2019-10-18
**Official Blind Review #1**

**Rating:** 3

**Review:**

The goal of the work is to quantify the dependency between contents in NLP. The method relies on parametrization of the joint probability of words in (2), and discussed some connections between the rank of the unfolded tensor T and the dependency level between two sets of words in a sentence.

The goal of this work is quite interesting, but the reviewer feel a bit challenging to follow the writing. The work seems to build upon a previous work, namely, tensor space language model (TSLM). But the paper does not introduce TSLM in detail, making the part relevant to TSLM quite inaccessible.

The analysis of the work is based on the model in (2), which is merely an approximation for the joint probability. This is fine, but maybe this point should be more spelled out in the paper.

The work also has an assumption that a naive Bayes model in (4) always holds for a set of w_1 ... w_n. This may need some more discussion and maybe a reference. Since the authors are considering a sequence, this may be related to de finetti's theorem and its extensions. But in that theorem many more assumptions are needed, eg., the RVs are exchangeable. Also, it is unknown if a finite K exists.

The proof of Claim 1 is a bit trivial, if one only considers one-hot encoding. In fact, the statement and proof of Claim 1 might be a bit loose. If one wishes that the SVD reveals the rank K, K has to be smaller than the outer dimensions of the tensor T. This was not specified in the statement.

The above also brings up another question: are the proofs all based on one-hot encoding of words? We know in NLP pre-trained word embeddings may be more useful. Do the proofs also apply to those cases, e.g., GloVe or Word2Vec?

Did all the experiments use one-hot embedding?

Overall, the reviewer feels that the work has an interesting motivation, and the goal is meaningful. The writing is a bit hard to access and the proofs might be a bit loose (did not check all of them. But Claim 1 is already a bit loose).



**Experience Assessment:**

I have read many papers in this area.

**Review Assessment: Checking Correctness Of Derivations And Theory:**

I assessed the sensibility of the derivations and theory.

**Review Assessment: Checking Correctness Of Experiments:**

I did not assess the experiments.

**Review Assessment: Thoroughness In Paper Reading:**

I read the paper at least twice and used my best judgement in assessing the paper.

---

> ### Author Response · Authors · 2019-11-15
> **Authors' response to reviewer 1**
>
> Thanks for your helpful advices. We have provided our responses below.
> 1. This work builds upon the previous work (namely tensor space language model, TSLM) which has been published and the detailed introduction of the TSLM will be added in Supplementary Appendices in the revised version.
> 2. In this work, we consider in Eq (4), the order of words cannot be exchanged. In Eq. (4), $k$ cannot be estimated. Therefore, we choose to estimate $k$ in the TSLM.
> 3. At present, the proof of Claim 1 only considers the one-hot coding. The case of a dense coding will be considered in the future work. Besides, in the experiments, we do not use the one-hot embedding and use the pre-trained word embeddings.
> 4. For the issue of writing, we will check and correct it.

---

### Official Review · AnonReviewer2 · 2019-10-21
**Official Blind Review #2**

**Rating:** 1

**Review:**

This paper first provides a theoretical interpretation of separation rank as a measure of a recurrent network's ability to capture contextual dependencies in natural language text. The analysis is primarily done in the context of tensor space language models (TSLM) that was demonstrated to be a generalisation of n-gram language models in earlier work. The theoretical derivations suggest that increasing a network's depth increases its separation rank exponentially, while increasing its width only increases its separation rank linearly. Based on this finding, the paper proposes a bidirectional adaptive recurrent neural network that adapts the network depth for each task using dynamic halting. Experiments on six NLP tasks demonstrate that increasing network depth helps more for tasks that generally require many long-range dependencies, while increasing network width is generally sufficient for tasks that require mostly short-term dependencies, although I disagree with the paper's dichotomy of tasks with long-term and short-term dependencies (see point 3 below).

Overall, this paper suffers from several serious issues, as listed below. I am therefore recommending a rating of "Reject" ahead of the authors' response.

1. This paper suffers from substantial clarity issues beyond simple grammatical errors. Some examples of the most serious clarity issues are as follows: (i) Section 6.1.1 lists "sentiment analysis" as one of the tasks, but Figure 2 has no entry for "sentiment analysis", yet instead features experimental results for "semantic classifier" which was never mentioned or introduced before; and (ii) Figure 2 shows "1st layer LSTM", "2nd layer LSTM", and "3rd layer LSTM", while (based on my reading) what the paper means are "1-layer LSTM", "2-layer LSTM", and "3-layer LSTM", hence highly confusing for the reader.

2. The proposed explanation about the proposed bidirectional adaptive RNN is really sparse, despite being a central part of the paper. The explanation of the proposed model is only contained in two paragraphs (Sections 5.1 and 5.2), and are not sufficient for the readers to understand the model. A more extensive explanation and intuition about what the model is like, and how it is similar or different to TSLM and standard RNNs, is required to improve this.

3. In the experiments, the paper assumes a false dichotomy of tasks that only require short-range dependencies (NER, POS tagging, and constituency parsing), and tasks that require long-range dependencies (WSD, sentiment analysis, and coreference resolution). This dichotomy is overly simplistic and ultimately false. For instance, constituency parsers often need to identify spans that are very long-distance in nature (for a recent investigation of this, see the work by Fried et al. (2019)), while sentiment analysis often only requires the model to identify a few salient words that are indicative of the sentiment, e.g. "excellent" or "terrible", hence not requiring much contextual dependencies.

4. Related to point 3, a better evaluation is to examine the cases that require long-range dependencies within each task, rather than assuming which tasks require long-range dependencies and which ones do not. An example of this is reporting e.g. constituency parsing performance for long-range spans and coreference resolution performance for long-distance entity chains.

5. The paper mostly fails to report performance comparison with existing numbers from prior work. For instance, the coreference performance (Fig. 2) are far below the result from Lee et al. (2017) that this paper is based on (Section 6.1.1), while the constituency parsing numbers (also Fig. 2) are also far below the reconciled span parser (Joshi et al., 2018) that this paper is also based on. This discrepancy calls into question the strength of the model implementation used in this paper.

6. Some missing citations, e.g. the use of adaptive computation time and dynamic halting in Universal Transformers (Dehghani et al., 2019).

7. This paper can benefit from careful copy-editing. Some examples of grammatical errors: (i) "Eq. 16" in Section 5.2 should be "Eq. 6", (ii) "to verify our theoretical, respectively" in Section 6 should be "to verify our theoretical [findings/derivations]", (iii) "ourself" on the caption of Table 1 should be "ourselves", (iv) "The model Ling et al. (2015) is used, which using bidirectional LSTMs" should be "The model [of] Ling et al. (2015) is used, which [used] bidirectional LSTMs", etc.

References
Daniel Fried, Nikita Kitaev, and Dan Klein. "Cross-domain generalization of neural constituency parsers". In Proc. of ACL 2019.

Kenton Lee, Luheng He, Mike Lewis, and Luke Zettlemoyer. "End-to-end neural coreference resolution". In Proc. of EMNLP 2017.

Vidur Joshi, Matthew E. Peters, and Mark Hopkins. "Extending a Parser to Distant Domains Using a Few Dozen Partially Annotated Examples". In Proc. of ACL 2018.

Mostafa Dehghani, Stephan Gouws, Oriol Vinyals, Jakob Uszkoreit, and Lukasz Kaiser. "Universal Transformers". In Proc. of ICLR 2019.


**Experience Assessment:**

I have read many papers in this area.

**Review Assessment: Checking Correctness Of Derivations And Theory:**

I assessed the sensibility of the derivations and theory.

**Review Assessment: Checking Correctness Of Experiments:**

I assessed the sensibility of the experiments.

**Review Assessment: Thoroughness In Paper Reading:**

I read the paper at least twice and used my best judgement in assessing the paper.

---

> ### Author Response · Authors · 2019-11-15
> **Authors' response to reviewer 2**
>
> Thanks very much for your detailed comments. Our responses are as follows.
> 1. For clarity issues and grammatical errors, we will check and correct them in this paper.
> 2. Regarding the relative less explanations of adaptive LSTM, we think that the main contributions are the theoretical analysis about the modeling ability of the contextual dependency, as well as the corresponding interpretable mechanism. Nevertheless, we will add more descriptions about the adaptive LSTM in the revised version of our paper.
> 3. For the problem of a dichotomy of tasks, this strategy is inspired by the recent work [2]. This work describes that for those tasks (e.g., NER, POS tagging, and constituency parsing), one needs the hidden vectors of the lower layer, to better capture short-range dependencies than those of the deeper layer.
> 4. For this work of the Universal Transformer [3], we will cite it in our paper.
> [1] Zhang L, Zhang P, Ma X, et al. A Generalized Language Model in Tensor Space[J]. arXiv preprint arXiv:1901.11167, 2019.
> [2] Matthew Peters, Mark Neumann, Luke Zettlemoyer, and Wen-tau Yih. Dissecting contextual word
> embeddings: Architecture and representation. In Proceedings of the 2018 Conference on Empirical
> Methods in Natural Language Processing, pp. 1499–1509, 2018a.
> [3] Mostafa Dehghani, Stephan Gouws, Oriol Vinyals, Jakob Uszkoreit, and Lukasz Kaiser. "Universal Transformers". In Proc. of ICLR 2019.

---

### Official Review · AnonReviewer3 · 2019-11-05
**Official Blind Review #3**

**Rating:** 3

**Review:**

This paper derives lower bounds on the separation rank of a wider class of recurrent NLP models in terms of its depth and number of hidden layers, demonstrating that both the number of hidden units as well as the number of layers improves the ability of NLP networks to model context dependency. It then introduces a novel bidirectional NLP variant that is supposed to capture a good trade-off between computational cost and performance.

The manuscript is very dense and does not follow a straight and easy-to-follow story line. In particular, the introduction of the bidirectional variant seems to substantially distract from the main story line of the paper (there is also no connection between the theoretical results to the bidirectional network). The improvements of the bidirectional models also seem to be minor, but no standard deviations for the performance results are reported.

A clear description as to which language models are captured by the TSLM model is missing. Also, it is unclear how tight the bounds actually are given that no value for m (the word length) is given. Finally, the title does not reflect the content of the paper (there is nothing interpretable about the network structure).

**Experience Assessment:**

I do not know much about this area.

**Review Assessment: Checking Correctness Of Derivations And Theory:**

I did not assess the derivations or theory.

**Review Assessment: Checking Correctness Of Experiments:**

I assessed the sensibility of the experiments.

**Review Assessment: Thoroughness In Paper Reading:**

I made a quick assessment of this paper.

---

> ### Author Response · Authors · 2019-11-15
> **Authors' response to reviewer 3**
>
> Thanks for your helpful comments. We have provided our responses below.
> 1. We actually provide some connections between the theoretical results and the bidirectional network. The theoretical results in Eq. (4) and Eq. (6) can guide the design of the bidirectional network. In Eq. (4), there is an important condition $\sum_{j=1 }^{K}p(y_j)=1$, which can help us design the dynamic halting. According to Eq. (6), we design the dynamic halting as $N(t)=min\{l:\sum_{i}^{l}p_{t}^{i} \leq {1-\epsilon} \} $ in Section 5.2.
> 2. (a) About the TSLM, this work has been published [1]. Inspired by this work, the separation rank in TSLM is proposed to measure the contextual dependency in our work. We have explained the background of TSLM in Section 2.2. We will add the detailed introduction of the TSLM in Supplementary Appendices.
> (b) For the title, “interpretable mechanism’’ means that we need to select the adaptive number of layers or the hidden units for the modeling of different sentences.

---

### Decision · Program_Chairs · 2019-12-19

**Decision:**

Reject

**Comment:**

This paper a theoretical interpretation of separation rank as a measure of a recurrent network's ability to capture contextual dependencies in text, and introduces a novel bidirectional NLP variant and tests it on several NLP tasks to verify their analysis.

Reviewer 3 found that the paper does not provide a clear description of the method and that a focus on single message would have worked better. Reviewer 2 made a claim of several shortcomings in the paper relating to lack of clarity, limited details on method, reliance on a 'false dichotomy', and failure to report performance. Reviewer 1 found the goals of the work to be interesting but that the paper was not clear, that the proofs were not rigorous enough, and clarity of experiments. The authors responded to the all the comments. The reviewers felt that their comments were still valid and did not adjust their ratings.

Overall, the paper is not yet ready in its current form. We hope that the authors will find valuable feedback for their ongoing research.